# Predictive value of red cell distribution width to albumin ratio for acute kidney injury in patients with acute pancreatitis

Jin Zhao, Yuan Peng, Chenyang Xu◉*

Intensive Care Unit, The First People's Hospital of Kunshan Affiliated with Jiangsu University, Kunshan, Jiangsu, China

* xcy8790@163.com

## Abstract

### Background

Red blood cell volume distribution width (RDW) and albumin have been increasingly recognized for their roles in inflammatory diseases. However, the relationship between RDW-to-albumin ratio (RAR) and acute kidney injury (AKI) in acute pancreatitis (AP) patients has not been clarified. Therefore, the research aims to explore the correlation between RAR and AKI risk in critically ill patients with AP.

### Methods

Patients diagnosed with AP from the Medical Information Mart for Intensive Care IV (MIMIC-IV) database were included in this retrospective study. The primary outcome measure was the incidence of AKI. Logistic regression analysis and restricted cubic spline were used to evaluate the relationship between RAR and AKI incidence in AP patients. Subgroup analyses were used to test interactions.

### Results

In total, 600 patients were enrolled in the study. The incidence of AKI was 77.3%. Based on multiple logistic regression analysis, RAR exhibited a positive association with AKI incidence as either a continuous (OR 1.46, 95% CI 1.22–1.76, P < 0.05) or categorical variable (OR 2.79, 95% CI 1.69–4.59, P < 0.05). The restricted cubic splines model illustrated the linear relationship between higher RAR and increased risk of AKI in AP patients.

### Conclusions

High RAR is an independent risk factor for AKI in critically ill patients with AP. Early assessment of RAR may facilitate risk stratification to guide clinical management, thereby improving clinical outcomes.

**Data availability statement:** All relevant data are within the manuscript and its Supporting Information files.

**Funding:** Kunshan Development Zone Medical and Health Science and Technology Innovation Project (X23-189-101591) to YP, and Special Project for Social Development Science and Technology of Kunshan City (X24-038-101326) to YP.

**Competing interests:** The authors have declared that no competing interests exist.

## Introduction

Acute pancreatitis (AP) is a necro-inflammatory disease of the pancreas characterized by abnormal activation of pancreatic enzymes, which triggers systemic inflammation and contributes to poor prognosis [1,2]. Acute kidney injury (AKI), a common complication in AP patients characterized by a rapid deterioration of renal function, further exacerbates AP-related adverse outcomes and is associated with high mortality [3]. The pathophysiology of AP-associated AKI (AP-AKI) is multifactorial, and recent studies indicate that oxidative stress is a key driver of AP-AKI. Excess reactive oxygen species (ROS) production during acute pancreatitis not only exacerbate inflammation and pancreatic tissue damage, but also induce endothelial dysfunction and renal hypoperfusion, resulting in the development of AKI. The inflammatory cascade triggered by AP causes further tissue damage, underlining the complex role of inflammation and oxidative stress in AP [4,5]. Notably, recent research highlights the crosstalk between pancreatic inflammation and renal dysfunction in AP. Pancreatic injury releases pro-inflammatory cytokines (e.g., TNF-α, IL-6) and damage-associated molecular patterns (DAMPs) into the systemic circulation. Then these factors trigger renal tubular apoptosis and impair renal microcirculation [6,7]. Conversely, compromised renal function exacerbates pancreatic injury via accumulated uremic toxins, reduced clearance of inflammatory mediators, and electrolyte disturbances [8]. This interaction emphasizes the need for biomarkers that reflect both local pancreatic inflammation and systemic organ crosstalk.

Red blood cell distribution width (RDW), a routine hematological parameter reflecting erythrocyte size variability, has been increasingly considered as a biomarker of systemic inflammation and oxidative stress in recent years. Elevated RDW is associated with impaired antioxidant capacity and exacerbated inflammatory status, potentially reflecting redox imbalance in AP patients [9,10]. In contrast, serum albumin, a negative acute-phase protein, decrease in inflammatory status due to increased vascular permeability and reduced hepatic synthetic capacity [11–13]. The decrease in albumin levels not only indicates nutrient depletion but also reflects a persistent inflammatory state, both of which are commonly observed in severe AP [14]. Therefore, the RDW-to-albumin ratio (RAR) integrates two pathophysiological mechanisms: oxidative stress-driven erythrocyte dysregulation and inflammation-associated hypoalbuminemia, serving as a novel composite biomarker for evaluating disease severity in AP patients with AKI.

Research on prognostic biomarkers for AP-AKI has increased in recent years.. Numerous biomarkers have been found to effectively predict AKI development in AP patients, such as neutrophil gelatinase-associated lipocalin (NGAL), β2-microglobulin (β2-MG), and cystatin C [15, 16]. Despite their high sensitivity and specificity, these novel biomarkers are limited by high costs and the need for dynamic monitoring. In contrast, RAR can be rapidly calculated based on routine hematological and biochemical parameters obtained at admission, making it both convenient and affordable. However, the predictive effect of RAR on AP-AKI remains unconfirmed. In view of the role of systemic inflammation and oxidative stress in multiple organ failure [17,18], we speculated that RAR might serve as an important and easily accessible biomarker for

AKI prediction and risk stratification. Thus, this study aims to explore the relationships between RAR and the incidence of AKI in AP patients, thereby providing new insights for early risk identification.

## Method

### Data sources

The data utilized in this study were derived from the Medical Information Mart for Intensive Care IV (MIMIC-IV) database, a publicly available critical care repository jointly developed by Massachusetts Institute of Technology researchers and clinical investigators at Beth Israel Deaconess Medical Center [19]. This database includes information on all patients admitted to the Beth Israel Deaconess Medical Center during the years from 2008 to 2019. Since all data were publicly available and de-identified, the study was exempt from ethical approval and informed consent requirements. The lead author of the study, Zhao Jin, completed the required credentialing process and obtained full access privileges to the MIMIC-IV database (User ID: 67544010).

### Data extraction

A total of 1151 patients diagnosed with AP were extracted according to the ICD codes 9 and 10. The exclusion criteria were as follows: age < 18 years; Length of hospital stay < 24h; patients with renal disease; and incomplete medical data. After applying the exclusion criteria, 600 patients were finally included in the study (**Fig 1**).

### Variable extraction

All variables were extracted from the MIMIC-IV database using PostgresSQL software. The demographic characteristics included sex, age and race. Laboratory variables within the first 24h of ICU admission included hemoglobin (Hb), red blood cell (Rbc), red blood cell distribution width (Rdw), hematocrit (Hct), white blood cell (Wbc), platelet count (Plt), albumin (Alb), prothrombin time (PT), international normalized ratio (INR), creatinine (Cr), blood urea nitrogen (Bun), aspartate aminotransferase (Ast) and alanine aminotransferase (Alt). The severity scores on admission included simplified acute physiology score II (SAPSII) and sequential organ failure assessment (SOFA) score. Comorbidities included sepsis, hypertension, diabetes, myocardial infarction (MI), chronic obstructive pulmonary disease (COPD), invasive ventilation,

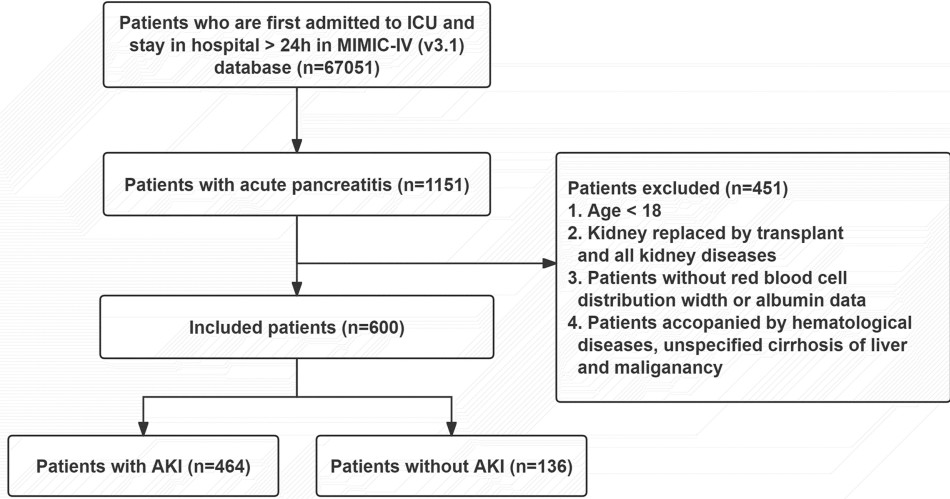

**Fig 1. Flow chart of patient selection.**

cardiopulmonary resuscitation (CPR) and the application of renal continue replacement therapy (CRRT). The defined endpoint event was the incidence of AKI resulted from AP.

## Statistical analysis

Continuous variables are presented as means with standard deviations, and categorical variables as counts with percentages. For continuous variables, continuous variables with normal distribution were compared by one-way ANOVA, while non-normally distributed variables were analyzed using Kruskal-Wallis test. The Chi-square or Fisher's exact test was used for categorical variables. Multiple logistic regression analysis was performed to calculate the Odds Ratio (OR) and the 95% confidence interval (CI) for the RAR and incidence of AKI within each subgroup. Model 1 was unadjusted; model 2 was adjusted for sex, age and race; and model 3 was additionally adjusted for Wbc, Hct, CRRT, map, creatinine, Bun and Ast. Restricted cubic spline (RCS) was performed to further validate the potential nonlinear relationships between RAR and the incidence of AKI. To verify the stability of our results, stratified analyses were conducted based on the following factors, including age, sex, sepsis, MI, CHF, COPD, hypertension, diabetes, invasivevent and RRT. Statistical analyses were performed with Empowerstats software (version 6.0). P-value < 0.05 was considered statistically significant.

# Result

## Baseline characteristics

According to the inclusion and exclusion criteria, 600 patients were included in the study, and the patient screening flow chart is shown in Fig 1. According to occurrence of AKI or not, the involved AP patients were divided into AKI group and non-AKI group. There were no significant differences in sex, race, HR, Wbc, Plt, Hb, MI, COPD, diabetes, and CPR between the groups. However, significant differences were observed in age, ICU time, hospital time, SOFA score, Map, Rdw, Rbc, Alb, Cr, Bun, PT, INR, Alt, Ast, RAR, sepsis, CHF, hypertension, demand for invasive ventilation, and CRRT requirement (P < 0.05, Table 1).

## Univariate and multivariate analyses of RAR in AP patients

According to univariate analysis, age, hospital time, ICU time, sepsis, CHF, hypertension, SAPS II, SOFA score, invasive ventilation, RDW, Cr, Alb, Bun, and RAR were shown to be the primary confounders which influenced AKI incidence (P < 0.05) (S1 Table). Based on multiple logistic regression analysis, RAR showed a positive correlation with AKI as either a continuous (OR 1.46, 95% CI 1.22–1.76, P < 0.05) or categorical variable (OR 2.79, 95% CI 1.69–4.59, P < 0.05, Table 2).

## ROC curve analysis

We analyzed the predictive value of RAR, RDW and albumin for AKI in AP patients using ROC curve analysis. The result showed that the area under the curve (AUC) of RAR was superior to those of RDW and Albumin (Fig 2). In addition, we analyzed differences of RAR in different AKI stages based on the KDIGO guideline. The results reflected that RAR in the AKI stage 3 was significantly higher than others (P < 0.05, S1 Fig).

## Restricted cubic spline

The RCS analysis was adjusted for the effects of sex, age, race, Wbc, CRRT, map, creatinine, Bun and Ast. It indicated a linear association between RAR and AKI after adjusting for confounding factors (P non-linear = 0.128, Fig 3).

## Subgroup analysis

Subgroup analysis was performed to assess the relationship between RAR and AKI among the various subgroups. No significant interaction effect was found in any subgroups after stratifying by sex, age, hypertension, DM, SOFA score, and severity of AP (all P for interaction > 0.05, Table 3).

**Table 1. Baseline characteristics of included patients grouped by the occurrence of AKI.**

| Characteristics | No-AKI (n = 136) | AKI (n = 464) | P-value |
|---|---|---|---|
| Age | 53.58 ± 16.72 | 59.37 ± 17.99 | <0.001 |
| Gender (Female) | 64 (47.06%) | 194 (41.81%) | 0.277 |
| Race | | | 0.079 |
| Asian | 7 (5.15%) | 11 (2.37%) | |
| Black | 15 (11.03%) | 30 (6.47%) | |
| White | 77 (56.62%) | 301 (64.87%) | |
| Others | 37 (27.21%) | 122 (26.29%) | |
| ICU time (d) | 2.06 (1.48-3.58) | 4.54 (2.27-13.46) | <0.001 |
| Hospital time (d) | 9.31 (5.64-14.88) | 16.50 (8.73-27.91) | <0.001 |
| SOFA | 4.00 (2.00-5.00) | 7.00 (4.00-11.00) | <0.001 |
| HR (tpm) | 104.16 ± 20.92 | 102.39 ± 21.48 | 0.466 |
| Map (mmHg) | 91.59 ± 18.49 | 86.82 ± 20.42 | 0.007 |
| Wbc (10^9/L) | 11.90 (8.73-16.50) | 12.90 (8.80-18.50) | 0.132 |
| Rdw (10^12/L) | 14.39 ± 1.54 | 14.87 ± 2.07 | 0.032 |
| Plt (10^9/L) | 190.50 (140.00-263.00) | 198.50 (146.00-280.75) | 0.554 |
| Rbc (10^12/L) | 3.74 ± 0.78 | 3.97 ± 0.90 | 0.019 |
| Alb (g/dL) | 3.13 ± 0.62 | 2.93 ± 0.68 | 0.004 |
| Cr (mg/dL) | 0.80 (0.60-1.20) | 1.10 (0.70-1.80) | <0.001 |
| Bun (mmol/dL) | 14.00 (8.75-23.25) | 20.50 (13.00-32.00) | <0.001 |
| PT (s) | 13.60 (12.20-14.90) | 4.20 (12.70-16.20) | 0.002 |
| INR | 1.20 (1.10-1.40) | 1.30 (1.10-1.50) | 0.005 |
| Alt (IU/L) | 36.00 (19.50-101.00) | 49.00 (25.00-159.00) | 0.013 |
| Ast (IU/L) | 45.00 (29.00-109.00) | 69.00 (33.00-167.50) | 0.009 |
| Hb (g/dL) | 11.66 ± 2.37 | 12.01 ± 2.54 | 0.202 |
| RAR | 4.79 ± 1.11 | 5.42 ± 1.75 | <0.001 |
| Sepsis | | | <0.001 |
| No | 75 (55.15%) | 119 (25.65%) | |
| Yes | 61 (44.85%) | 345 (74.35%) | |
| MI | | | 0.119 |
| No | 127 (93.38%) | 412 (88.79%) | |
| Yes | 9 (6.62%) | 52 (11.21%) | |
| CHF | | | <0.001 |
| No | 128 (94.12%) | 367 (79.09%) | |
| Yes | 8 (5.88%) | 97 (20.91%) | |
| COPD | | | 0.386 |
| No | 113 (83.09%) | 370 (79.74%) | |
| Yes | 23 (16.91%) | 94 (20.26%) | |
| Hypertension | | | 0.033 |
| No | 58 (42.65%) | 246 (53.02%) | |
| Yes | 78 (57.35%) | 218 (46.98%) | |
| Diabetes | | | 0.748 |
| No | 93 (68.38%) | 324 (69.83%) | |
| Yes | 43 (31.62%) | 140 (30.17%) | |
| Invasive Ventilation | | | <0.001 |
| No | 105 (77.21%) | 191 (41.16%) | |
| Yes | 31 (22.79%) | 273 (58.84%) | |

*(Continued)*

**Table 1.** (Continued)

| Characteristics | No-AKI (n=136) | AKI (n=464) | P-value |
|---|---|---|---|
| CPR | | | 0.886 |
| No | 135 (99.26%) | 460 (99.14%) | |
| Yes | 1 (0.74%) | 4 (0.86%) | |
| CRRT | | | <0.001 |
| No | 136 (100.00%) | 387 (83.41%) | |
| Yes | 0 (0.00%) | 77 (16.59%) | |

**Table 2. Multiple logistic regression analysis between RAR and AKI.**

| Variable | Model1 | | Model2 | | Model3 | |
|---|---|---|---|---|---|---|
| | OR (95%) | P value | OR (95%) | P value | OR (95%) | P value |
| RAR | 1.35 (1.16, 1.57) | <0.0001 | 1.36 (1.17, 1.59) | <0.0001 | 1.46 (1.22, 1.76) | <0.0001 |
| RAR, tertiles | | | | | | |
| <=5.43 | Reference | | Reference | | Reference | |
| >5.43 | 2.42 (1.56, 3.74) | <0.0001 | 2.45 (1.57, 3.83) | <0.0001 | 2.79 (1.69, 4.59) | <0.0001 |

Model1: Unadjusted.

Model2: adjusted for Gender, Age, Race.

Model3: adjusted for Gender, Age, Race, Wbc; Hct; CRRT; Map; Creatinine; Bun; Ast.

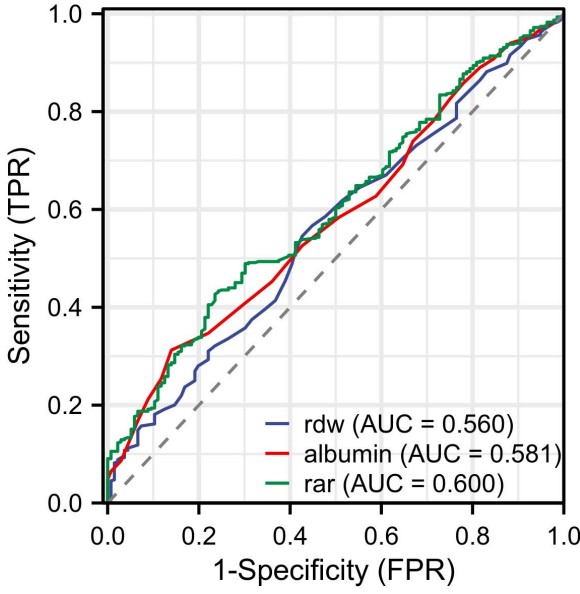

**Fig 2. ROC curves for biomarkers to predict AKI.** ROC curves of RAR (green line), Rdw (blue line), albumin (red line). RAR, red blood cell distribution width-to-albumin ratio; Rdw, red blood cell distribution.

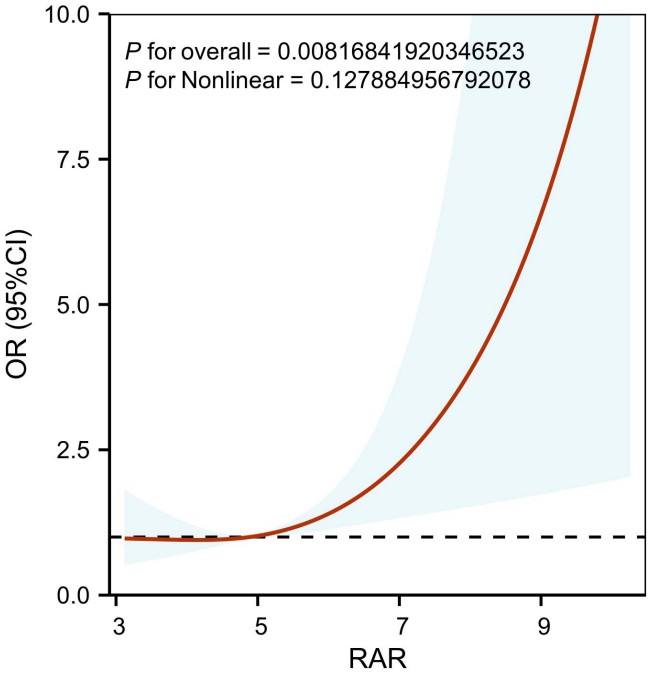

**Fig 3. Restricted cubic spline curve of RAR with acute kidney injury incidence.** RAR was entered as a continuous variable. Hazard ratios were adjusted for Gender, Age, Race, Wbc, Hct, CRRT, Map, Creatinine, Bun, Ast. RAR, red blood cell distribution width-to-albumin ratio; WBC, white blood cell; Hct, hematocrit; CRRT, continuous renal replacement therapy; MAP, mean arterial pressure; BUN, blood urea nitrogen; AST, aspartate aminotransferase. The red line represents the estimated values. The shaded area represents the corresponding 95% confidence intervals.

## Discussion

This study examined the association between RAR levels and the incidence of AKI in patients with AP. The result revealed that elevated RAR was associated with an increased risk of AKI in critically ill AP patients. The correlation was statistically significant even after adjusting for potential confounding factors. The RCS model revealed a linear association between RAR and AKI risk in patients with AP. Subgroup analyses and interaction tests revealed stable relationship between RAR and AKI in various subgroups.

Oxidative stress and inflammation play a pivotal role in the pathogenesis of pancreatitis and the development of AP-related complications [20,21]. RDW is the coefficient of variation of RBC volume and represents RBC size heterogeneity [22]. Elevated RDW reflects severe systemic inflammatory response and high oxidative stress. Systemic inflammatory response during the course of severe AP can suppress erythrocytes maturation by interfering with the erythrocyte membrane and accelerate the migration of reticulocytes into the peripheral circulation, thereby leading to elevated RDW [23]. In addition, inflammatory cytokines inhibit iron metabolism and promote red blood cell apoptosis [24]. Notably, recent studies suggest that elevated RDW is associated with poor prognosis in AP patients [25–27]. Multiple organ failure in severe AP is the main factor leading to early death [28]. Serum albumin, which is the most abundant protein in human extracellular fluid, plays important physiological roles as a negative acute phase inflammatory biomarker and is widely used for assessing the inflammatory and nutritional condition of the human body. In the inflammatory condition, hypoproteinemia reflects impaired liver synthetic function, protein loss due to capillary leakage, and accelerated albumin catabolism [29]. Currently, albumin has been proposed as a reliable prognostic factor for critically ill patients [30]. Recent studies have shown that hypoalbuminemia is associated with poor prognosis of AP patients [31].

**Table 3. Subgroup analyses for the association between RAR and AKI.**

| AKI | N | OR | 95%CI | P value | P for interaction |
|---|---|---|---|---|---|
| Gender | | | | | 0.2758 |
| Female | 258 | 1.50 | 1.19, 1.90 | 0.0006 | |
| Male | 342 | 1.27 | 1.04, 1.55 | 0.0215 | |
| Age | | | | | 0.3259 |
| < 45 | 200 | 1.50 | 1.16, 1.95 | 0.0023 | |
| 45-71.8 | 200 | 1.32 | 1.04, 1.69 | 0.0234 | |
| > 72 | 200 | 1.12 | 0.85, 1.47 | 0.4119 | |
| Sepsis | | | | | 0.3763 |
| No | 194 | 1.38 | 1.05, 1.81 | 0.0207 | |
| Yes | 406 | 1.19 | 0.99, 1.43 | 0.0657 | |
| MI | | | | | 0.7749 |
| No | 539 | 1.36 | 1.16, 1.60 | 0.0001 | |
| Yes | 61 | 1.26 | 0.77, 2.07 | 0.3594 | |
| CHF | | | | | 0.1827 |
| No | 495 | 1.41 | 1.19, 1.66 | <0.0001 | |
| Yes | 105 | 1.02 | 0.68, 1.51 | 0.9401 | |
| COPD | | | | | 0.9257 |
| No | 483 | 1.35 | 1.15, 1.60 | 0.0004 | |
| Yes | 117 | 1.33 | 0.93, 1.90 | 0.1189 | |
| Hypertension | | | | | 0.7165 |
| No | 304 | 1.30 | 1.05, 1.61 | 0.0142 | |
| Yes | 296 | 1.38 | 1.11, 1.71 | 0.0038 | |
| Diabetes | | | | | 0.3514 |
| No | 417 | 1.30 | 1.09, 1.54 | 0.0030 | |
| Yes | 183 | 1.53 | 1.12, 2.10 | 0.0077 | |
| Invasive Vent | | | | | 0.3811 |
| No | 296 | 1.30 | 1.07, 1.59 | 0.0098 | |
| Yes | 304 | 1.13 | 0.89, 1.43 | 0.3134 | |
| CRRT | | | | | 0.6893 |
| No | 505 | 1.29 | 1.10, 1.51 | 0.0015 | |
| Yes | 95 | 1.03 | 0.39, 2.72 | 0.9478 | |

As a novel composite biomarker combining RDW and albumin levels, RAR has been extensively investigated in various inflammation-associated diseases. There are several studies that have suggested that RAR is an independent prognostic indicator for patients with sepsis and is associated with a poor clinical prognosis [32,33]. Simultaneously, higher RAR levels were significantly associated with increased mortality in patients with acute respiratory distress syndrome (ARDS) [34]. Notably, recent studies have demonstrated a significant association between RAR and the incidence of contrast-induced acute kidney injury. Notably, recent studies have demonstrated a significant association between RAR and the incidence of contrast-induced acute kidney injury (CI-AKI) [35]. Moreover, the strong link between CI-AKI and inflammatory pathogenesis has been widely recognized in clinical research [36,37]. However, no studies have reported an association between RAR and AKI in patients with AP. Therefore, we hypothesized that RAR is associated with the development of AKI in AP. Our research revealed that elevated RAR in AP patients was correlated with a higher AKI incidence. Notably, ROC curve analysis demonstrated that RAR showed a higher AUC than either RDW or albumin alone for predicting AKI in patients with AP, and this superior AUC directly indicated its stronger discriminatory ability and clinical significance.

## Strengths and limitations

Our study has several strengths. Firstly, the study is the first to identify an independent association between RAR and AKI risk in patients with AP. Composite biomarkers provide a more comprehensive view of systemic inflammatory responses and metabolic dysregulation compared to conventional single-parameter biomarkers. Secondly, the study is also the first to establish a linear relationship between RAR and the occurrence of AKI based on the RCS model. Thirdly, the study demonstrated robust consistency in the association between RAR and AKI through subgroup analysis and interaction tests after adjustment for confounders, highlighting its potential value as a predictive factor in AP-AKI.

However, this study also has several limitations. Firstly, this was a single center retrospective study with an inherent bias in data collection. Some unmeasured confounding factors, including patients' detailed medication history and the exact onset time of AP, may potentially affect the accuracy of the study results. Second, the data used in this study was derived from a single database, which may limit the generalizability of the study findings. Thirdly, we only measured RDW and serum albumin levels at ICU admission and did not investigate their temporal trends, which could have provided more detailed insights into disease progression. Fourthly, factors with important clinical significance, such as fluid resuscitation volume within the first 24 hours, intra-abdominal pressure, and AKI onset timing relative to ICU admission, could not be included in the study due to data missing in the database. Finally, our findings demonstrate a significant association between RAR and the incidence of AP-AKI. Importantly, all the findings reflect an association rather than causation. These results should be interpreted as exploratory and necessitate further validation through rigorously designed multicenter prospective studies.

## Conclusion

High RAR serves as an independent predictor for AKI in AP patients. Early assessment of RAR may facilitate risk stratification to guide clinical management, thereby improving clinical outcomes.

## Supporting information

**S1 Table. Univariate Logistic Regression Analysis for AKI.**
(DOCX)

**S1 Fig. The RAR levels in different stages based on the KDIGO guideline.**
(TIFF)

## Author contributions

**Conceptualization:** Chenyang Xu.

**Data curation:** Jin Zhao, Yuan Peng, Chenyang Xu.

**Investigation:** Jin Zhao, Yuan Peng.

**Methodology:** Chenyang Xu.

**Resources:** Yuan Peng.

**Software:** Jin Zhao.

**Supervision:** Jin Zhao, Yuan Peng, Chenyang Xu.

**Validation:** Jin Zhao, Yuan Peng, Chenyang Xu.

**Visualization:** Yuan Peng.

**Writing – original draft:** Jin Zhao, Chenyang Xu.

**Writing – review & editing:** Jin Zhao, Yuan Peng, Chenyang Xu.

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
