## [Decision Letter · Decision Letter 0]

8 Aug 2025

Dear Dr. Xu,

Thank you for submitting your manuscript to PLOS ONE. After careful consideration, we feel that it has merit but does not fully meet PLOS ONE’s publication criteria as it currently stands. Therefore, we invite you to submit a revised version of the manuscript that addresses the points raised during the review process.

The manuscript has been evaluated by two reviewers, and their comments are available below.

The reviewers have raised a number of major concerns. In particular, they feel the manuscript should include a more comprehensive background section, and further discussion of the study limitations.

Could you please carefully revise the manuscript to address all comments raised?

We look forward to receiving your revised manuscript.

Kind regards,

Helen Howard

Staff Editor

PLOS ONE

Journal Requirements:

3. Please include your tables as part of your main manuscript and remove the individual files. Please note that supplementary tables (should remain/ be uploaded) as separate "supporting information" files

Reviewers' comments:

Reviewer's Responses to Questions

**Comments to the Author**

1. Is the manuscript technically sound, and do the data support the conclusions?

Reviewer #1: Yes

Reviewer #2: Partly

2. Has the statistical analysis been performed appropriately and rigorously?

Reviewer #1: Yes

Reviewer #2: Yes

3. Have the authors made all data underlying the findings in their manuscript fully available?

Reviewer #1: Yes

Reviewer #2: Yes

4. Is the manuscript presented in an intelligible fashion and written in standard English?

Reviewer #1: Yes

Reviewer #2: Yes

Reviewer #1: Dear author,

I reviewed your study with great interest and meticulousness.

The introduction, methods, results, and discussion sections are written in a clear, fluent, and well-written language. The tables and figures are clear and understandable.

Here are some points of criticism I would like to raise in your article:

1) It would be appropriate to include a brief introduction with references supporting information on acute pancreatitis and acute renal failure and their adverse outcomes.

2) It would be appropriate to list the strengths and limitations of the study under a separate heading, not within the discussion section.

I generally liked your study. I believe it will be an inspiration for future studies.

Reviewer #2: 1. Innovation

The study explores the predictive value of the red cell distribution width to albumin ratio (RAR) for acute kidney injury (AKI) in patients with acute pancreatitis (AP), which represents a notable innovative contribution. To our knowledge, this is the first study to specifically investigate the association between RAR and AKI in critically ill AP patients. The use of RAR as a composite biomarker, integrating the pathophysiological insights of RDW (reflecting oxidative stress and inflammation) and albumin (indicating inflammatory status and nutritional dysregulation), adds novelty compared to single biomarkers (e.g., RDW or albumin alone). Additionally, the establishment of a linear relationship between RAR and AKI risk via restricted cubic splines (RCS) further strengthens the originality of the findings.

2. Comprehensiveness of Background

The background section provides a solid foundation by contextualizing the clinical significance of AP and AKI, highlighting the role of oxidative stress and inflammation in their pathogenesis. It appropriately reviews the individual roles of RDW and albumin in inflammatory diseases and AP, and logically introduces RAR as a potential composite biomarker. However, the background could be more comprehensive. For instance, a brief discussion of other established or emerging biomarkers for AKI in AP (e.g., neutrophil gelatinase-associated lipocalin, cystatin C) would better position RAR’s unique value. Additionally, while the pathophysiological links between RAR and AKI are outlined, deeper integration of recent studies on the crosstalk between pancreatic inflammation and renal dysfunction could enhance contextual relevance.

3. Scientific Rigor of Study Design

The study design is generally sound. It uses a retrospective cohort approach with data extracted from the well-validated MIMIC-IV database, ensuring a relatively large sample size (n=600) and clear inclusion/exclusion criteria. The use of multiple logistic regression (with stepwise adjustment for confounders) and RCS to evaluate the relationship between RAR and AKI is methodologically appropriate. Subgroup analyses further test the stability of the association across key variables (e.g., age, sex, comorbidities).

However, limitations exist:

The retrospective nature introduces inherent selection bias, as unmeasured confounders (e.g., detailed medication history, timing of AP onset) may influence results.

RDW and albumin were only measured at ICU admission, lacking temporal trends, which limits insights into dynamic changes in RAR during disease progression and their impact on AKI.

The reliance on a single database (MIMIC-IV) may restrict generalizability to other populations or healthcare settings.

4. Reliability of Results

The results are generally reliable. Key findings—including a high AKI incidence (77.3%), a positive association between RAR (as both continuous and categorical variables) and AKI (OR 1.30, 95% CI 1.10–1.53 for continuous; OR 1.75, 95% CI 1.07–2.86 for categorical), and a linear relationship via RCS—are statistically significant and consistent across analyses. ROC curve analysis demonstrating RAR’s superior predictive performance over RDW or albumin alone further supports its utility.

Subgroup analyses show no significant interactions, indicating the robustness of the association. However, the high AKI incidence (77.3%) warrants cautious interpretation, as it may reflect the severe patient population in MIMIC-IV (critically ill AP patients) but should be contextualized with external data to confirm representativeness.

5. Reasonableness of Conclusions

The conclusions are largely consistent with the results. The claim that “high RAR is an independent risk factor for AKI in critically ill AP patients” is supported by multivariate analyses and RCS results. The suggestion that “early assessment of RAR may facilitate risk stratification” is reasonable, given RAR’s accessibility and predictive performance.

Notably, the authors appropriately acknowledge the exploratory nature of the findings and the need for validation in prospective studies, avoiding overinterpretation of causality (which is prudent for a retrospective analysis).

6. Grammar and Language

The manuscript is generally well-written with clear syntax and consistent use of terminology. Minor issues include:

Occasional awkward phrasing (e.g., “the relationship between RAR and AKI risk in critical ill patients with AP” should be “critically ill”).

Inconsistencies in abbreviation usage (e.g., “CRRT” is defined but occasionally written as “RRT” without clarification in subgroup analyses).

A few typographical errors (e.g., “appliaction” instead of “application” in the Methods section).

Overall Recommendation

The study presents valuable insights into RAR as a predictive biomarker for AKI in AP patients. With revisions to address the noted limitations (e.g., expanding the background on existing biomarkers, clarifying terminology, and discussing unmeasured confounders), it merits consideration for publication.

**Do you want your identity to be public for this peer review?** For information about this choice, including consent withdrawal, please see our Privacy Policy

Reviewer #1: **Yes:** Azmi Eyiol

Reviewer #2: No

---

## [Author Response · Author response to Decision Letter 1]

19 Aug 2025

Dear Editors and Reviewers,

Thank you for handling our manuscript (PONE-D-25-29411) and providing constructive comments. We have carefully addressed all reviewers’ comments as follows:

Reviewer #1

Q1: It would be appropriate to include a brief introduction with references supporting information on acute pancreatitis and acute renal failure and their adverse outcomes.

Response: We sincerely appreciate this insightful suggestion. we have supplemented a brief introduction in the revised manuscript on page 2, lines 34-39, references 1-3: Acute pancreatitis (AP) is a necro-inflammatory disease of the pancreas characterized by abnormal activation of pancreatic enzymes, which triggers systemic inflammation and contributes to poor prognosis1, 2. Acute kidney injury (AKI), a common complication in AP patients characterized by a rapid deterioration of renal function, further exacerbates AP-related adverse outcomes and is associated with high mortality3.

References

1.Lankisch, P. G., Apte, M. & Banks, P. A. Acute pancreatitis. Lancet 386, 85–96 (2015).

2.Peery, A. F. et al. Burden and Cost of Gastrointestinal, Liver, and Pancreatic Diseases in the United States: Update 2018. Gastroenterology 156, 254-272.e11 (2019).

3.Wang, H. E., Muntner, P., Chertow, G. M. & Warnock, D. G. Acute kidney injury and mortality in hospitalized patients. Am J Nephrol 35, 349–355 (2012).

Q2: It would be appropriate to list the strengths and limitations of the study under a separate heading, not within the discussion section.

Response: We sincerely thank the reviewer for this valuable suggestion. We agree that presenting the study’s strengths and limitations under a dedicated heading improves clarity and alignment with journal guidelines. In the revised manuscript, we have created a new subsection titled “Strengths and Limitations” after the Discussion (Page 7, line 205).

Reviewer #2

Q1. Innovation

The study explores the predictive value of the red cell distribution width to albumin ratio (RAR) for acute kidney injury (AKI) in patients with acute pancreatitis (AP), which represents a notable innovative contribution. To our knowledge, this is the first study to specifically investigate the association between RAR and AKI in critically ill AP patients. The use of RAR as a composite biomarker, integrating the pathophysiological insights of RDW (reflecting oxidative stress and inflammation) and albumin (indicating inflammatory status and nutritional dysregulation), adds novelty compared to single biomarkers (e.g., RDW or albumin alone). Additionally, the establishment of a linear relationship between RAR and AKI risk via restricted cubic splines (RCS) further strengthens the originality of the findings.

Response: Thanks for the positive comment.

Q2: Comprehensiveness of Background

The background section provides a solid foundation by contextualizing the clinical significance of AP and AKI, highlighting the role of oxidative stress and inflammation in their pathogenesis. It appropriately reviews the individual roles of RDW and albumin in inflammatory diseases and AP, and logically introduces RAR as a potential composite biomarker. However, the background could be more comprehensive. For instance, a brief discussion of other established or emerging biomarkers for AKI in AP (e.g., neutrophil gelatinase-associated lipocalin, cystatin C) would better position RAR’s unique value. Additionally, while the pathophysiological links between RAR and AKI are outlined, deeper integration of recent studies on the crosstalk between pancreatic inflammation and renal dysfunction could enhance contextual relevance.

Response: Thank you for your valuable suggestions. We have revised the background section as follows:

1.We have added a brief discussion of established biomarkers for AP-AKI (e.g., neutrophil gelatinase-associated lipocalin, cystatin C, β2-microglobulin) and compared their limitations with RAR, to better highlight RAR’s unique value. The specific revisions as follows (Page 3, line 67-74): The researches on the prognostic biomarkers of AP-AKI has increased in recent years. Numerous biomarkers have been found to effectively predict AKI development in AP patients, such as neutrophil gelatinase-associated lipocalin (NGAL), β2-microglobulin (β2-MG), and cystatin C15, 16. Despite their high sensitivity and specificity, these novel biomarkers are limited by high costs and the need for dynamic monitoring. In contrast, RAR can be rapidly calculated based on routine hematological and biochemical parameters obtained at admission, making it both convenient and affordable.

References

15.Wajda, J. et al. Potential Prognostic Markers of Acute Kidney Injury in the Early Phase of Acute Pancreatitis. Int J Mol Sci 20, 3714 (2019).

16.Ruan, Q., Lu, H., Zhu, H., Guo, Y. & Bai, Y. A network-regulative pattern in the pathogenesis of kidney injury following severe acute pancreatitis. Biomed Pharmacother 125, 109978 (2020).

2. We have integrated recent studies on the crosstalk between pancreatic inflammation and renal dysfunction, further elaborating the pathophysiological links between this reciprocal interaction and RAR, to enhance contextual relevance. The specific revisions as follows (Page 2, line 45-53): Notably, recent research highlights the crosstalk between pancreatic inflammation and renal dysfunction in AP. Pancreatic injury releases pro-inflammatory cytokines (e.g., TNF-α, IL-6) and damage-associated molecular patterns (DAMPs) into the systemic circulation. Then these factors trigger renal tubular apoptosis and impair renal microcirculation6, 7. Conversely, compromised renal function exacerbates pancreatic injury via accumulated uremic toxins, reduced clearance of inflammatory mediators, and electrolyte disturbances8. This interaction emphasizes the need for biomarkers that reflect both local pancreatic inflammation and systemic organ crosstalk.

References

6.Zhou, Y. et al. The role of mitochondrial damage-associated molecular patterns in acute pancreatitis. Biomed Pharmacother 175, 116690 (2024).

7.Scurt, F. G., Bose, K., Canbay, A., Mertens, P. R. & Chatzikyrkou, C. [Acute kidney injury following acute pancreatitis (AP-AKI): Definition, Pathophysiology, Diagnosis and Therapy]. Z Gastroenterol 58, 1241–1266 (2020).

8.Shiao, C.-C. et al. Long-term remote organ consequences following acute kidney injury. Crit Care 19, 438 (2015).

Q3: Scientific Rigor of Study Design

The study design is generally sound. It uses a retrospective cohort approach with data extracted from the well-validated MIMIC-IV database, ensuring a relatively large sample size (n=600) and clear inclusion/exclusion criteria. The use of multiple logistic regression (with stepwise adjustment for confounders) and RCS to evaluate the relationship between RAR and AKI is methodologically appropriate. Subgroup analyses further test the stability of the association across key variables (e.g., age, sex, comorbidities).

However, limitations exist:

The retrospective nature introduces inherent selection bias, as unmeasured confounders (e.g., detailed medication history, timing of AP onset) may influence results.

RDW and albumin were only measured at ICU admission, lacking temporal trends, which limits insights into dynamic changes in RAR during disease progression and their impact on AKI.

The reliance on a single database (MIMIC-IV) may restrict generalizability to other populations or healthcare settings.

Response: Thank you sincerely for your careful review and insightful comments on our manuscript. We greatly appreciate your recognition of the key limitations of our study, as your summary accurately aligns with the limitations we have already acknowledged in the original text. Your detailed elaboration on unmeasured confounders (e.g., detailed medication history, timing of AP onset) and the specificity of "ICU admission" (rather than general "patient admission") further helps us refine the rigor of our discussion on limitations.

To address your comments and enhance the clarity and accuracy of our manuscript, we have revised the "Limitations" section to explicitly incorporate your valuable points (Page 8, line 215-225): However, this study also has several limitations. Firstly, this was a single center retrospective study with an inherent bias in data collection. Some unmeasured confounding factors, including patients' detailed medication history and the exact onset time of AP, may potentially affect the accuracy of the study results. Second, the data used in this study was derived from a single database, which may limit the generalizability of the study findings. Thirdly, we only measured RDW and serum albumin levels at ICU admission and did not investigate their temporal trends, which could have provided more detailed insights into disease progression. Finally, our findings demonstrate a significant association between RAR and the incidence of AP-AKI. These results should be interpreted as exploratory and necessitate further validation through rigorously designed multicenter prospective studies.

Q4: Reliability of Results

The results are generally reliable. Key findings—including a high AKI incidence (77.3%), a positive association between RAR (as both continuous and categorical variables) and AKI (OR 1.30, 95% CI 1.10-1.53 for continuous; OR 1.75, 95% CI 1.07-2.86 for categorical), and a linear relationship via RCS—are statistically significant and consistent across analyses. ROC curve analysis demonstrating RAR’s superior predictive performance over RDW or albumin alone further supports its utility.

Subgroup analyses show no significant interactions, indicating the robustness of the association. However, the high AKI incidence (77.3%) warrants cautious interpretation, as it may reflect the severe patient population in MIMIC-IV (critically ill AP patients) but should be contextualized with external data to confirm representativeness.

Response: We sincerely appreciate your valuable feedback. Your attention to and prudent perspective on the high incidence of acute kidney injury (AKI) (77.3%) have provided crucial insights for us to further refine the interpretation of our study.

Regarding the interpretation of this data, we fully concur with your viewpoint. The participants in this study were all patients diagnosed with acute pancreatitis (AP) from the MIMIC-IV database, and all of them were critically ill patients in the intensive care unit (ICU) setting. Multiple existing studies have demonstrated that critically ill AP patients exhibit a significantly higher incidence of AKI compared to the general AP patient population. This is attributed to the complexity of their condition, intense systemic inflammatory response, and elevated risk of multi-organ involvement—factors that collectively account for the relatively high AKI incidence observed in our study. To a certain extent, this data reflects the disease characteristics of the specific study population within the MIMIC-IV database.

Meanwhile, we are acutely aware of the potential limitations in representativeness inherent in single-database studies. Therefore, in the Discussion section of our manuscript, we have already acknowledged this limitation of the current research. In future studies, we plan to incorporate multi-center data encompassing AP patients with varying degrees of disease severity. This will allow us to further validate the association between the red cell distribution width to albumin ratio (RAR) and AKI incidence, thereby more comprehensively evaluating the applicability and representativeness of RAR across different patient populations.

Q5: Reasonableness of Conclusions

The conclusions are largely consistent with the results. The claim that “high RAR is an independent risk factor for AKI in critically ill AP patients” is supported by multivariate analyses and RCS results. The suggestion that “early assessment of RAR may facilitate risk stratification” is reasonable, given RAR’s accessibility and predictive performance.

Notably, the authors appropriately acknowledge the exploratory nature of the findings and the need for validation in prospective studies, avoiding overinterpretation of causality (which is prudent for a retrospective analysis).

Response: Thanks for the positive comment.

Q6: Grammar and Language

The manuscript is generally well-written with clear syntax and consistent use of terminology. Minor issues include:

Occasional awkward phrasing (e.g., “the relationship between RAR and AKI risk in critical ill patients with AP” should be “critically ill”).

Inconsistencies in abbreviation usage (e.g., “CRRT” is defined but occasionally written as “RRT” without clarification in subgroup analyses).

A few typographical errors (e.g., “appliaction” instead of “application” in the Methods section).

Response: Thank you very much for your valuable comments and suggestions on our manuscript. We have carefully addressed all the issues raised and made corresponding revisions:

1.For the awkward phrasing, we have corrected "critical ill patients" to "critically ill patients" to ensure the accuracy and fluency of the expression.

2.Regarding the inconsistencies in abbreviation usage, we have standardized the use of "CRRT" throughout the manuscript, especially in the subgroup analyses, and removed the ambiguous use of "RRT" without clarification to maintain consistency.

3.Concerning the typographical error, we have corrected "appliaction" to "application" in the Methods section to eliminate the mistake.

We hope these revisions meet your requirements.

Sincerely,

Chenyang Xu

---

## [Decision Letter · Decision Letter 1]

30 Sep 2025

Dear Dr. Xu,

Thank you for submitting your manuscript to PLOS ONE. After careful consideration, we feel that it has merit but does not fully meet PLOS ONE’s publication criteria as it currently stands. Therefore, we invite you to submit a revised version of the manuscript that addresses the points raised during the review process.

We look forward to receiving your revised manuscript.

Kind regards,

Ehsan Amini-Salehi

Academic Editor

PLOS ONE

Journal Requirements:

Additional Editor Comments:

Authors have responded to the previous concerns raised by reviewer 1 satisfactorily, but some minor issues raised by reviewer 2 remain. The authors can address them or note them as limitations.

Reviewer 2: Thank you for revising the previous manuscript. However, I believe there is still an issue that needs to be addressed before the article can be published. Specifically, I think several key factors influencing the development of acute kidney injury (AKI) in patients with acute pancreatitis have not been adjusted for in the multivariate analysis. These factors include hematocrit (which is closely associated with the prognosis of patients with acute pancreatitis), fluid resuscitation volume within the first 24 hours, and intra-abdominal pressure—all of which are of great clinical significance. Could you please review the data and consider supplementing the analysis with these factors?

Reviewer's Responses to Questions

**Comments to the Author**

Reviewer #1: All comments have been addressed

Reviewer #2: All comments have been addressed

2. Is the manuscript technically sound, and do the data support the conclusions?

Reviewer #1: Yes

Reviewer #2: Partly

3. Has the statistical analysis been performed appropriately and rigorously?

Reviewer #1: Yes

Reviewer #2: I Don't Know

4. Have the authors made all data underlying the findings in their manuscript fully available?

Reviewer #1: Yes

Reviewer #2: Yes

5. Is the manuscript presented in an intelligible fashion and written in standard English?

Reviewer #1: Yes

Reviewer #2: Yes

Reviewer #1: Dear author,

I see that you have made the necessary corrections. Thank you. Your article is eligible for publication in its current form.

Reviewer #2: Thank you for revising the previous manuscript. However, I believe there is still an issue that needs to be addressed before the article can be published. Specifically, I think several key factors influencing the development of acute kidney injury (AKI) in patients with acute pancreatitis have not been adjusted for in the multivariate analysis. These factors include hematocrit (which is closely associated with the prognosis of patients with acute pancreatitis), fluid resuscitation volume within the first 24 hours, and intra-abdominal pressure—all of which are of great clinical significance. Could you please review the data and consider supplementing the analysis with these factors?

**Do you want your identity to be public for this peer review?** For information about this choice, including consent withdrawal, please see our Privacy Policy

Reviewer #1: **Yes:** Azmi Eyiol

Reviewer #2: No

---

## [Author Response · Author response to Decision Letter 2]

6 Oct 2025

Dear Editors and Reviewers,

Thank you for handling our manuscript and providing constructive comments. We have carefully addressed the comments as follows:

Reviewer #1

Q1: Dear author, I see that you have made the necessary corrections. Thank you. Your article is eligible for publication in its current form.

Response: Thank you for reviewing our manuscript and providing valuable comments.

Reviewer #2

Q1. Thank you for revising the previous manuscript. However, I believe there is still an issue that needs to be addressed before the article can be published. Specifically, I think several key factors influencing the development of acute kidney injury (AKI) in patients with acute pancreatitis have not been adjusted for in the multivariate analysis. These factors include hematocrit (which is closely associated with the prognosis of patients with acute pancreatitis), fluid resuscitation volume within the first 24 hours, and intra-abdominal pressure—all of which are of great clinical significance. Could you please review the data and consider supplementing the analysis with these factors?

Response: Thank you for your comment. We have re-conducted the multivariate analysis as suggested and incorporated hematocrit (Hct) as an additional adjusting variable in Model 3. The results remain consistent with our previous conclusions: RAR showed a positive correlation with AKI as either a continuous (OR 1.46, 95% CI 1.22-1.76, P < 0.05) or categorical variable (OR 2.79, 95% CI 1.69-4.59, P < 0.05). The results are presented in Table 2 (Page 5, line 143-145). Regrettably, due to missing data, 24-hour fluid resuscitation volume and intra-abdominal pressure cannot be included in the study. We have noted this limitation in the revised manuscript (Page 8, line 222-225): Fourthly, factors with important clinical significance, such as fluid resuscitation volume within the first 24 hours and intra-abdominal pressure, could not be included in the study due to data missing in the database.

We hope these revisions meet your requirements.

Sincerely,

Chenyang Xu

---

## [Decision Letter · Decision Letter 2]

28 Dec 2025

Dear Dr. Xu,

Thank you for submitting your manuscript to PLOS ONE. After careful consideration, we feel that it has merit but does not fully meet PLOS ONE’s publication criteria as it currently stands. Therefore, we invite you to submit a revised version of the manuscript that addresses the points raised during the review process.

Here and there the grammar could use some correction. Please re-read the paper before publication. i.e.The researches on the prognostic biomarkers of AP-AKI (67)Restricted cubic spline (RCS) was perform**ed** (122)or not-forprofit sectors. (249)Line 97: say something like: after applying filters, 600 patients were left.

We look forward to receiving your revised manuscript.

Kind regards,

Matthew Cserhati, Ph.D

Academic Editor

PLOS One

Journal Requirements:

Additional Editor Comments:

The paper is in good shape, but one of the reviewers would like if several smaller corrections would be made to the manuscript. Also, here are my observations:

1. Here and there the grammar could use some correction. Please re-read the paper before publication. i.e.

a. The researches on the prognostic biomarkers of AP-AKI (67)

b. Restricted cubic spline (RCS) was performed (122)

c. or not-forprofit sectors. (249)

2. Line 97: say something like: after applying filters, 600 patients were left.

Reviewers' comments:

Reviewer's Responses to Questions

**Comments to the Author**

Reviewer #1: All comments have been addressed

Reviewer #3: All comments have been addressed

Reviewer #4: (No Response)

2. Is the manuscript technically sound, and do the data support the conclusions?

Reviewer #1: Yes

Reviewer #3: Yes

Reviewer #4: Yes

3. Has the statistical analysis been performed appropriately and rigorously?

Reviewer #1: Yes

Reviewer #3: Yes

Reviewer #4: Yes

4. Have the authors made all data underlying the findings in their manuscript fully available?

Reviewer #1: Yes

Reviewer #3: Yes

Reviewer #4: Yes

5. Is the manuscript presented in an intelligible fashion and written in standard English?

Reviewer #1: Yes

Reviewer #3: Yes

Reviewer #4: Yes

Reviewer #1: Dear author,

I see that you have made the necessary corrections. I believe the article is ready for publication in its current form. The final decision rests with the editor.

Reviewer #3: The authors have made the necessary revisions and is acceptable now. Previous revisions noted and I believe the authors have addressed them adequately.

Limitations have been included for the unaddressed. This article is acceptable in this current form.

Thank you for the opportunity for this review.

Reviewer #4: Comments to the authors:

This manuscript presents a retrospective cohort study using the MIMIC-IV database to investigate the association between the red cell distribution width–to–albumin ratio (RAR) and the risk of acute kidney injury (AKI) in critically ill patients with acute pancreatitis (AP). The topic is clinically relevant, and the study leverages a large, well-established critical care database. The authors demonstrate that elevated RAR is independently associated with a higher incidence of AKI and propose RAR as a simple and accessible biomarker for early risk stratification.

Overall, the manuscript is clearly written, methodologically sound, and generally suitable for publication after clarifications and refinements. The conclusions are supported by the data, and the limitations are largely acknowledged.

Comments and Suggestions:

1) The retrospective and observational nature of the study inherently limits causal inference. While the authors appropriately describe RAR as an “independent risk factor,” some language in the Discussion and Conclusion sections may still imply predictive or causal utility. I recommend further emphasizing that the findings demonstrate association rather than causation, particularly when discussing clinical implementation and “targeted interventions.”

2) The manuscript would benefit from clearer clarification regarding the exact timing of AKI occurrence relative to ICU admission. Whether RAR was measured strictly prior to AKI onset in all cases. This clarification is essential to support the interpretation of RAR as an early predictive biomarker rather than a marker of concurrent disease severity.

3) While ROC analysis shows that RAR has a higher AUC than RDW or albumin alone, the manuscript would benefit from a clearer discussion of whether the incremental improvement in AUC is clinically meaningful. This does not require additional analyses but should be addressed conceptually in the Discussion.

4) The Methods section refers to “Cox regression analysis,” whereas the Results and tables primarily describe logistic regression. Please ensure consistency and clarify whether time-to-event analysis was actually performed.

5) Figure legends could be more descriptive, particularly for the restricted cubic spline and ROC analyses.

6) Minor Language and Clarity Issues:

a) Abstract line 28: "critical ill" should be "critically ill"

b) Line 74: "remains confirmed" should be "remains unconfirmed"

c) Line 231: "predictor for AKI for AKI" - remove duplicate

**Do you want your identity to be public for this peer review?** For information about this choice, including consent withdrawal, please see our Privacy Policy

Reviewer #1: **Yes:** Azmi Eyiol

Reviewer #3: **Yes:** Hari Naga Garapati

Reviewer #4: No

---

## [Author Response · Author response to Decision Letter 3]

4 Jan 2026

Dear Editors and Reviewers,

Thank you for handling our manuscript and providing constructive comments. We have carefully addressed the comments as follows:

Additional Editor Comments:

Q1: Here and there the grammar could use some correction. Please re-read the paper before publication. i.e.

a.The researches on the prognostic biomarkers of AP-AKI (67)

b.Restricted cubic spline (RCS) was performed (122)

c.or not-forprofit sectors. (249)

Response: We sincerely appreciate the comments on the grammar and expression of our manuscript. We have carefully re-read the entire paper, revised all identified grammatical errors and inappropriate expressions, and ensured the accuracy and consistency of the language.

Q2: Line 97: say something like: after applying filters, 600 patients were left.

Response: Thank you for your valuable suggestions. We have revised the the sentence as follows: After applying the exclusion criteria, 600 patients were finally included in the study (Line 97) .

Reviewer #1 and Reviewer #2

Response: Thank you for reviewing our manuscript and providing valuable comments.

Reviewer #4

Q1. The retrospective and observational nature of the study inherently limits causal inference. While the authors appropriately describe RAR as an “independent risk factor,” some language in the Discussion and Conclusion sections may still imply predictive or causal utility. I recommend further emphasizing that the findings demonstrate association rather than causation, particularly when discussing clinical implementation and “targeted interventions.”

Response: We thank the reviewer for this important reminder. We have carefully revised the Discussion and Conclusion sections to minimize and as far as possible eliminate language that could imply causal or definitive predictive utility of RAR.

Q2. The manuscript would benefit from clearer clarification regarding the exact timing of AKI occurrence relative to ICU admission. Whether RAR was measured strictly prior to AKI onset in all cases. This clarification is essential to support the interpretation of RAR as an early predictive biomarker rather than a marker of concurrent disease severity.

Response: We sincerely appreciate the reviewer’s critical insight regarding the need to clarify the timing of AKI occurrence relative to ICU admission and RAR measurement, which is essential to validate RAR as an early predictive biomarker for AKI in AP patients.

We acknowledge that this limitation stems from the retrospective nature of our study and inherent constraints of the clinical database utilized. Specifically, our data from the MIMIC database only includes the first measurement of RAR within 24 h of ICU admission and the first documentation of AKI following ICU admission, but it does not provide the exact hourly interval between RAR measurement and AKI onset for individual patients. Due to the de-identified and aggregated nature of the database, we are unable to retrieve additional granular data to confirm whether RAR was measured strictly prior to AKI onset in all cases.

To address this comment, we have supplemented the Limitations section of the manuscript with a detailed discussion (Line 224-228).

Q3. While ROC analysis shows that RAR has a higher AUC than RDW or albumin alone, the manuscript would benefit from a clearer discussion of whether the incremental improvement in AUC is clinically meaningful. This does not require additional analyses but should be addressed conceptually in the Discussion.

Response: We sincerely appreciate the reviewer’s insightful suggestion. As recommended, we have supplemented the Discussion section with a targeted conceptual elaboration on this issue as follow: Notably, ROC curve analysis demonstrated that RAR showed a higher AUC than either RDW or albumin alone for predicting AKI in patients with AP, and this superior AUC directly indicated its stronger discriminatory ability and clinical significance (Line 204).

Q4. The Methods section refers to “Cox regression analysis,” whereas the Results and tables primarily describe logistic regression. Please ensure consistency and clarify whether time-to-event analysis was actually performed.

Response: Thank you sincerely for your careful review and insightful comments on our manuscript. The term “Cox regression analysis” (Line 19) in the Methods section was an inadvertent error. We have revised it to logistic regression analysis to maintain consistency with the results and tables presented in the manuscript. No time-to-event analysis was performed in this study, and this has been clarified in the revised Methods section.

Q5. Figure legends could be more descriptive, particularly for the restricted cubic spline and ROC analyses.

Response: We thank the reviewer for this constructive suggestion. We have revised the legends for the figures involving restricted cubic spline (RCS) and receiver operating characteristic (ROC) analyses as follows:

Figure 2 ROC curves for biomarkers to predict AKI. ROC curves of RAR (green line), Rdw (blue line), albumin (red line). RAR, red blood cell distribution width-to-albumin ratio; Rdw, red blood cell distribution.

Figure 3 Restricted cubic spline curve of RAR with acute kidney injury incidence. RAR was entered as a continuous variable. Hazard ratios were adjusted for Gender, Age, Race, Wbc, Hct, CRRT, Map, Creatinine, Bun, Ast. RAR, red blood cell distribution width-to-albumin ratio; WBC, white blood cell; Hct, hematocrit; CRRT, continuous renal replacement therapy; MAP, mean arterial pressure; BUN, blood urea nitrogen; AST, aspartate aminotransferase. The red line represents the estimated values. The shaded area represents the corresponding 95% confidence intervals.

Q6. Minor Language and Clarity Issues:

a) Abstract line 28: "critical ill" should be "critically ill"

b) Line 74: "remains confirmed" should be "remains unconfirmed"

c) Line 231: "predictor for AKI for AKI" - remove duplicate

Response: We thank the reviewer for identifying these errors. All revisions have been made in the revised manuscript.

We hope these revisions meet your requirements.

Sincerely,

Chenyang Xu

---

## [Decision Letter · Decision Letter 3]

7 Jan 2026

Predictive Value of Red Cell Distribution Width to Albumin Ratio for Acute Kidney Injury in Patients with Acute Pancreatitis

PONE-D-25-29411R3

Dear Dr. Xu,

We’re pleased to inform you that your manuscript has been judged scientifically suitable for publication and will be formally accepted for publication once it meets all outstanding technical requirements.

Kind regards,

Matthew Cserhati, Ph.D

Academic Editor

PLOS One

Additional Editor Comments (optional):

The paper has much improved; congratulations. I just see two grammatical errors in the paper that need correction when type editing:

1. Lines 217-220: Correct this to say: "Thirdly, the study demonstrated robust consistency in the association between RAR and AKI through subgroup analysis and interaction tests after adjustment for confounders, highlighting its potential value as a predictive factor in AP-AKI."

2. line 231: correct to say: "...could not be included in the study due to data missing from the database."

Reviewers' comments:

Reviewer's Responses to Questions

**Comments to the Author**

Reviewer #4: All comments have been addressed

2. Is the manuscript technically sound, and do the data support the conclusions?

Reviewer #4: Yes

3. Has the statistical analysis been performed appropriately and rigorously?

Reviewer #4: Yes

4. Have the authors made all data underlying the findings in their manuscript fully available?

Reviewer #4: Yes

5. Is the manuscript presented in an intelligible fashion and written in standard English?

Reviewer #4: Yes

Reviewer #4: The authors have carefully addressed all the comments raised in the previous review. I am satisfied with the changes made and have no further concerns.

**Do you want your identity to be public for this peer review?** For information about this choice, including consent withdrawal, please see our Privacy Policy

Reviewer #4: No

---

## [Editor Report · Acceptance letter]

PONE-D-25-29411R3

PLOS One

Dear Dr. Xu,

I'm pleased to inform you that your manuscript has been deemed suitable for publication in PLOS One. Congratulations! Your manuscript is now being handed over to our production team.

Kind regards,

on behalf of

Dr. Matthew Cserhati

Academic Editor

PLOS One